# Heritage Tourism and Nation-Building: Politics of the Production of Chinese National Identity at the Mausoleum of Yellow Emperor

Hongni Wei [1], Yi Yu [2] and Zhenjie Yuan [3,4,*]

1   School of Business, Guangdong University of Foreign Studies, Guangzhou 510006, China;
    nini2022@gdufs.edu.cn
2   School of Geography and Planning, Sun Yat-Sen University, Guangzhou 510275, China;
    yuyi39@mail.sysu.edu.cn
3   Centre for Human Geography and Urban Development, School of Geography and Remote Sensing,
    Guangzhou University, Guangzhou 510006, China
4   Guangdong Provincial Center for Urban and Migration Studies, Guangzhou 510006, China
*   Correspondence: zjyuan@gzhu.edu.cn

**Abstract:** As an important embodiment and carrier of Chinese traditional culture, the rituals and ceremonies in heritage tourism not only carry profound spiritual and cultural connotations, such as respecting nature and worshiping ancestors, they also enable people to gain a sense of identity. Therefore, this paper aims to explore the relationship between heritage tourism and the politics of identity-building from the perspective of critical toponymy based on the case study of the Mausoleum of the Yellow Emperor. Drawing on five years' of fieldwork at the Mausoleum of the Yellow Emperor in Shaanxi Province, China, this paper unfolds how heritage tourism has evolved into a certain kind of political agenda and a social engineering of nation-building. Relying on in-depth interviews and R language text analysis, this paper examines how Chinese national identity is produced, performed, and established through landscape naming and ritual performance in heritage tourism. This paper finds that (1) the naming and interpretation of ancestral roots cultural landscapes, as well as ritual performance in heritage tourism, are closely associated with cultural representations and reproductions of national identity. (2) The naming and interpretation of landscapes, as well as the performance of ritual practices in heritage tourism, are closely associated with national history and mythology. The tourists' behaviors and emotions, as well as their performance and experiences during the ritual practice at the heritage tourism site, help to construct national identity. The cultural production and invention are combined with actions enacted by governments and local residents, as well as visitors from different backgrounds. (3) The mausoleum creates cultural links for Taiwanese tourists to understand their ancestral roots culture and thus to enhance their connection with the mainland. This paper tries to explore the relationship between heritage tourism and identity-building from the perspective of critical toponymy, which has implications for place branding and marketing projects when promoting ancestral roots culture and place-named tourism. This research not only helps the individuals to realize and reappreciate the value of traditional culture and heritage, it also motivates the individuals to rethink their responsibilities in cultural inheritance and the innovative development of culture. It also helps to enhance the consciousness of the people on both sides of the strait as a community of the Chinese nation, as well as to promote the peaceful development of cross-strait relations.

**Keywords:** heritage; tourism; identity; landscape; ritual practice



## 1. Introduction

Scholars have observed the relationship between tourism and regional politics [1–12]. Specifically, heritage tourism is closely intertwined with nation-building, where national

heritage sites, such as monuments, palaces, and historical areas, are actively employed to promote the idea of nation [13]. The existing literature has investigated why and how heritage represents social identities; it has looked especially at the implicit function that heritage institutions play in representing and enhancing national identity [14–16]. However, national identity formation in the context of heritage tourism has largely been ignored [13].

Heritage is not just a tangible site; it is also the intangible embodiment, full of symbolic meanings [17,18]. Heritage serves not only as the representation of the attributes of a national culture but also as a kind of symbolic embodiment [19]. People construct and express their national identity through heritage, which creates connections among landscapes, historical events, national symbols, and rituals [20]. However, how those landscapes, national symbols, and rituals represent a nation and influence people's perceptions of a nation deserves further exploration [21].

This paper explores the relationships and the nexus between landscapes, rituals, and national identity, based on the case study conducted at the Mausoleum of the 'Yellow Emperor' in Shaanxi Province, China. The Chinese people have worshipped the Yellow Emperor (Xuan Yuan) as their ancestor since ancient times. Since the inception of the 'Reform and Opening-up' policy in 1978, an increasing number of Chinese—including people from Hong Kong, Macau, and Taiwan, as well as overseas Chinese—have been to the mausoleum in search of their ancestral roots. The mausoleum has become the most popular tourist destination for ancestor worship. The mausoleum provides an opportunity to investigate how China (as a nation) and Chinese (as a national identity) are represented, performed, and constructed through landscape naming and the visitors' ritual experience in heritage tourism. Furthermore, it offers a chance to understand how heritage tourism has engaged in the relations between mainland China and Taiwan.

## 2. Literature Review

### 2.1. Heritage Tourism and National Identity

According to a recent report issued by the International Council on Monuments and Sites (ICOMOS), cultural heritage and sustainable development are highly entangled [22]. The principal of "no cultural, no future" indicates that culture and cultural heritage should be integrated in both short-term recovery and long-term development strategies in response to the goal of cultural and community resilience in the UN 2030 Agenda. Indeed, under the background of globalization, local culture and identity may face complex and overlapping threats [23]. Therefore, governments and organizations are expected to implement cultural heritage plans to reinforce the value of traditional culture, thus nurturing attachment and a sense of belonging to heritage and the places in which it is located [24,25]. As early as 2003, the Convention for the Protection of the Intangible Cultural Heritage (ICHC) was adopted by UNESCO 2003; it took community and environment as the core components of tangible and intangible cultural heritage. For instance, intangible cultural heritage, such as beliefs, views, crafts, rituals, and other traditional practices, is deeply rooted in the history of local communities [26]. Inheriting those traditions and skills from the past is an effective approach for the local community to maintain cultural continuity and to reinforce identity.

In consideration of the nexus between heritage, culture, and nation-building, McKay puts forward his often-cited concept of the "tourism state" and argues that it becomes a world-making player by manipulating the symbolic image of the province [27]. Tourism, especially heritage tourism, plays a central role in developing, promoting, and strengthening national identities among domestic and even international tourism [28]. Park regards heritage as a kind of cultural production and explores its fundamental role in maintaining national unity, where heritage tourism is associated with experiencing both the material and the psychosocial remnants of the nation's past [13]. It plays a significant role not only in preserving and reconstructing the collective memory of a social group but also in the maintaining and promoting of the national development [13,29].

It is worth noting that there is still no consensus on the definition of national identity [30]. For Kohn, nationalism is a state of mind and consciousness which requires the

individual's supreme loyalty to the state [31]. Anderson pointed out that nations are invented as imaginary communities [32]. For Renan, the nation is a spiritual principle that presupposes a shared past and legacy of memories [30]. According to Geertz, national identity is an accumulation of sentiments, including assumed blood ties, race, ethnicity, language, religion, and customs, which help the individuals to identify themselves as belonging to a particular nation [33]. In a broader sense, Liu argues that national identity can be roughly divided into three dimensions: ethnic, cultural, and institutional identity, referring to an individual's perception of her/his connections to an ethnic group, a spectrum of culture, and a social or political entity [34]. Whatever criteria are employed to define national identity, scholars seem to agree that the idea of 'nation' is tied to a sense of history—to the memories, traditions, and sentiments that have been handed down from one generation to the next. Here, national identity is understood as an organic formation, where innate cultural characteristics are deeply grounded, and thus, cultural heritage is not only part of the history of a nation, it also plays a significant role in representing, shaping, and constructing a national identity.

### 2.2. Landscape and Nation-Building

The tourism landscape is different from the ordinary cultural landscape because of its symbolic connotations [35]. Tourism landscapes derive their symbolic meaning from a specific historical context. Thus, understanding heritage tourism involves understanding and interpretating the connotations behind it. Questions such as those which ask how symbolic landscapes become a container of history and how and why they are preserved attract scholarly attention.

As Meinig has pointed out, 'every mature nation has its symbolic landscapes' [36]. Those symbolic landscapes are powerfully evocative because they are closely related to the history and collective memory of the nation and the national identity. Palmer believes that some specific landscapes become the dominant symbols of the nation, arguing that landscapes are quite powerful because they indicate the social construction of a nation and the individual position in the nation [37]. Heritage tourism has been denoted as the essential codes which signify the national history and culture [38]. The heritage site not only serves as the mirror of the collective past of the nation but also plays a central role in the recovery of the heritage industry [39].

The granting of tangible and intangible heritage brings not only opportunities but also challenges to local governance. With regard to opportunities, the place might be promoted as a destination of national or universal value which would attract more enterprises to invest in local tourism and sustainable development [40]. As for challenges, a balance between economic development and the protection of cultural diversity is hard to achieve [41]. In his recent publication, Porter introduces a strategic planning and place branding of the English Lake District in the UK in the UNESCO World Heritage Sites (WHS) [42]. He argues that the nomination and management of WHS cultural landscapes involve an action of "making" (i.e., planning and branding), in which the articulation of a place's universal value is crucial [43–45]. In this sense, heritage governance needs to balance the relationship between place, culture, and identities and the interests of different stakeholders.

Indeed, heritage, as a form of landscape, is the scene of action and an expression of human ideas, thoughts, beliefs, and feelings. The understanding of landscape is the essential vehicle for nation-building because it usually symbolizes the nation's history and the roots, thereby engaging the people in a relationship with their past that can help them to make sense of the present [29]. Anderson believes that nation is an imagined community whereby each individual 'imagines' that his or her fellow citizens have the same basic understanding of what the nation is all about. National identity is experienced through communication with friends and relatives rather than through interactions with the entire nation [29]. It can be clearly understood that nation-building mainly refers to national construction to describe the greater integration of state and society because citizenship brings loyalty and belonging to a modern nation state. However, how to successfully

maintain such loyalty in order to maintain long-term stability becomes a problem, which attracts the researchers' attention. The key to state construction lies not in democratization, but in the inclusive political integration established by the exchange of power between the state and the citizens [46]. Nation-building in this article refers to the process and the result of using state power to construct national identity. Smith argues that in Western societies these connections are also embodied in the historical memories, myths, symbols, and rituals that reflect the different paths that the Western world has taken towards national consciousness [47]. These symbols and ceremonies include the distinctive features of nations, the flags, anthems, processions, mythology, architecture, etc. It is through these concepts that the nation is made visible and unique to its own members, including how individuals understand and experience through heritage tourism—especially symbolic landscapes of heritage—in order to become a member of a particular nation [29].

In this vein, the landscape of heritage—both in the material and the symbolic forms— is not merely a physical expression of boundaries but an expression of its past, present, and future [47]. A group of cultural geographers represented by Cosgrove focuses on the symbolic significance of landscape, stating that people endow space, place, and landscape with symbolic meaning [48]. National symbols are used to draw boundaries and to show the uniqueness of the nation [49,50]. As Anderson pointed out, nations originate in the past and the landscape in a nation can evoke affections which show the identity of those who live within the borders [30]. Palmer once pointed out that people can identify their belonging to the nation through heritage tourism [29]. Emotional sentiment, which is also expressed as a kind of belonging and togetherness, helps to maintain and develop collective consciousness [51,52]. Emotional sentiment is related with the symbols of the group and community. Hobsbawm argues that symbols play a very significant role in nation-building because they indicate a sense of belonging [53]. Smith believes that the visualization of symbols emphasizes the basic concept of national belonging [47].

### 2.3. Toponymy and Politics of Place Naming

Toponymy is the study of place names; in traditional toponymy research, geographers play the role of collecting and analyzing the temporal and spatial patterns of toponymy, taking place names as a clue to explore the changes of natural, social, and historical environments. However, this descriptive research paradigm left geographical name research on the edge of the field of geography for a long time [54]. Secondly, the traditional perspective focuses on the naming of objects and the linguistic or historical attributes of place names, which ignores the cultural politics and even the interpersonal conflicts behind place names. Since the 1980s, there has been a wave of a critical turn in the study of geographical names abroad, especially in the Western geography community. Traditionally, most place names are combined with the local environment, and they are mostly based on the characteristics of the local geographical environment. In the Western world, the basis for naming and renaming is associated with political meanings. Since the 18th century, it has become quite common in Europe to use street names to celebrate and commemorate historical events or figures in national history. At this point, the naming of place names began to be clearly separated from the local natural and cultural history and turned to being closely related to the nation as well as its politics. Therefore, when the government changes the old place names and assigns them new ones, the process of place name change is no longer suitable for understanding and explanation simply through "natural succession", which is the social origin of the Western critical toponymy [55].

Influenced by the postmodern structural criticism of geography, critical toponymy began to rise in Western countries. Alderman, Azaryahu, and Rose-Redwood were the pioneers who began to discuss the political, cultural, and symbolic meaning of place names [56–60]. How the political awareness and social values affect the place names has also attracted researchers' attention [61]. Alderman analyzes the influence of civic leaders during the naming process, based on the streets and roads named after Martin Luther King [62]. Azaryahu believes that street names can be the typical political symbol

for showing power and social order [63]. Rose-Redwood explores the hidden meaning behind place naming through the study of New York's street names. He believes the place-naming process is closely related to the construction of place attachment and collective memory. In addition, the negotiation of different actors during the naming process also needs attention [64–67]. In the 21st century, Chinese scholars also began to pay attention to the political and symbolic meaning of place names. Liu and Zhu pointed out that local politics and the cultural and commercial activities are the factors that influence the naming of squares [68]. According to historical evolution of place names in Taipei, Huang ( finds that the naming rights of places are constantly dominated by the authority [69]. Feng et al. analyzes the influence of local and national colonial power, based on the naming of Chinese urban streets from the perspective of Gramsci's theory of hegemony [70].

Critical toponymy has focused less on place names and more on the cultural and political aspects of naming [60]. Place names are not negative spatial designators but active actors in the practice of place construction. The traditional paradigm occupies a dominant position in the research of place names in China, and there is a lack of breakthroughs in the research methods and perspectives on modern geographical names [71]. The explanations and reasons for the dramatic changes of the geographical name landscape under the background of modernity and cultural globalization still need further research. The critical turn of toponymy will give domestic researchers more insight and inspiration. Previous scholars have focused more on discussing the associations between place names and economic capital and political power, as well as social identity. Recent researchers tried to discuss the negotiations of cultural and political meanings behind place names [72]. Yet, there are still few studies on the political and cultural significance in terms of the naming of places and landscapes. The politics behind place naming still needs further research.

Spatial assemblages and their evolution are the critical issues in toponymic studies [73]. In Bourdieu's theory of symbolic rights, symbols have the power to shape and strengthen the social order and the political system [74].In the process of ideological games and political struggles, the authority constructs a system of unique symbols which are commonly recognized by members of the society in order to make distinctions among different groups. Place names form under the political struggle and convey symbolic power. They are used to convey the symbolic meaning of "self" to "the other". Therefore, place names refer to the psychological features and the political ideology of the ruling class. Not only the authority but also the public has the naming right. Thus, place names not only serve as important carriers to construct collective memory and place attachment but also act as mirrors that reflect peoples' livelihoods [72]. A growing number of scholars have emphasized the importance of understanding place naming as a contested spatial practice rather than viewing place names as transparent signifiers that designate places as 'objects' or 'artifacts' within a predefined geographical space [59]. Place names serve as a kind of social agency to explore the cultural and political meanings in the naming process, while for the cultural landscape, they are an extensional form of representation and a series of symbolic systems full of meanings [75]. Representation here refers to the external things that truly express the human being's feelings and emotions [76]. In order to better understand landscapes, scholars need to observe the social and political process of place naming behind cultural landscapes. Therefore, this paper aims to discuss the political and cultural meanings, as well as the cultural politics in the process of landscape naming and ritual performance through heritage tourism.

Geographers have a long history of studying tourism and toponymy, but seldom discuss the relationship between tourism and toponymy. Light points out that it is not only the names of people that attract tourists, place names can also be attractions [77]. Place names have significance for tourists. MacCannell identifies naming as part of the process of defining tourist attractions. He points out the concept of the 'sight of sacralization' [78]. Allocating the name of a place is the first stage of sight sacralization; place names play a crucial role in signifying a tourist site. Such names suggest the distinction, originality, and authenticity of different places. Most of the studies from the perspective of history

and geography focus on the evolution of place names during different historical periods. The perspective of critical toponymies has become the new approach to studying the relationship between tourism and place names [59]. Critical toponymy regards place names as cultural symbols participating in the process of place construction, which not only focuses on the cultural and political implications behind place names but also emphasizes the explanations of the reasons behind the changes of place names [60].

Alderman has discussed the relationship between place naming, commemoration, and identity building [56]. Light has discussed the relationship between toponymy and tourism as well as the consumption of place names and landscapes [77]. Landscapes are endowed with sacred meanings and are conceived of as symbols of collective memory and identity [48,79]. Memorial ritual is not only the reflection of social changes but also the transformation of cultural and symbolic meanings invested by the current world [80]. The local society focuses more on the symbolic significance of mausoleums and tombs as tombs and mausoleums are not only the focal points of identity and the expressions of relationships with the land but are also the core part of the practice of memorial ceremonies. At the same time, mausoleums are designed as spacious parklands for tourism, leisure, and entertainment [81]. In a traditional Chinese rural village, the remains of ancestors were placed into a landscape of mausoleums and tombs, which is also called a kind of ritual territoriality [82]. Existing research suggests that the landscape of mausoleums and tombs is associated with cultural politics, which means that sacred time and space are reconceptualized, and rituals are reinvented to suit the conditions of modernity [83]. Within which, the practices of invention (i.e., the naming and interpretation of landscapes, ritual processes and their interpretations, etc.) involving complex negotiations and resistance stem from the dilemmas between the dominant and the subordinate, the traditional and the modern, as well as the majority and minority that deserve further studies.

*2.4. Nation-Building through Landscape Naming and Ritual Experience*

It is expected that tangible and physical elements of heritage, such as the establishing and naming of statues, squares, and monuments, will contribute to the construction of national identity. Political subjectivities become territorialized through spatial legitimacy and dominance imprinted on materials and representational practices. Political meanings, ideologies, and memories are inscribed onto the landscapes and ritual processes within which a regime can establish spatial legitimacy and domination. Light has examined the relationship between place naming and tourism consumption [77]. Researchers have developed critical accounts of how the naming of places and landscapes becomes a form of cultural politics [84,85]. The meaning of the landscape is intertwined with the politics, economy, and culture of different social situations. Together, they are constantly constructed, deconstructed, and reemphasized in the social and cultural changes. The history, development, and evolution of landscapes are also the history of Chinese politics. The material landscape, the power relations behind landscapes, and the negotiation of the meaning of landscapes construct a relatively consistent discourse [86]. The material landscape and the display of landscapes reflect the relationship between economy, politics, culture, and power. The diachronic evolution of landscapes emphasizes the role of landscapes in cultural and social relations from the political dimension [87]. Place names are related not only to the cultural and historical geographies of landscapes but also to the political geographies of power and dissent [88]. They argue that the naming of places can be viewed as a metaphor related to the contextual combinations of discourses, actors, institutions, material objects, and place-naming processes [84].

Indeed, place names and the interpretation of landscapes help to create symbols of national memories and ideologies. They can evoke powerful images and connotations, contributing to the development of a sense of national consciousness. Scholars contend that the naming of landscapes and ritual performance are a geopolitical practice by which the state can foster a sense of nationhood [89]. Ritual is the behavior interpretation of mythological narratives, which can stimulate and enhance people's differentiation between

self and others. Landscapes, collective memories, and other cultural symbols help to create a commemorative atmosphere. Through repetitive rituals, individuals make connections with the nation. National identity is a kind of feeling and is how the perception of the participants attending the ritual can co-build a sense of national consciousness [90]. Memory and emotions are stimulated by these kinds of repetitive rituals, which help to strengthen national cohesion. Ritual helps to create a sense of community in social and political life [91]. Connerton believes that ritual performance and commemorative ceremonies are conducive to the construction of collective memory, thereby stimulating a sense of national consciousness and national identity [92]. By assigning nationalistic meanings and commemorative significance to places (i.e., monuments and mausoleums), a regime creates a territorial reality that can develop national identities [93,94]. For example, various studies have focused on how the place inscriptions of colonial power transformed the everyday landscape of the colonial themes. Rao examined how the place inscription of British politicians on the Hong Kong cityscape led to the territorialization of colonial power [95]. Nash highlighted how the use of British royal names in Ireland became 'the reminder of Elizabethan expansion' [67]. Most of these researchers agree that the making of a place is indeed a reproduction of the cultural and geopolitical relationships between groups, boundaries, and territories across time and spaces [94].

Mausoleums, cemeteries, and other deathscapes are subject to national regulations and influenced by local history and cultural norms. How these kinds of old traditions were invented in the modern society needs academic attention. To figure out these kind of questions, we try to discuss how national image is constructed through the naming and interpretation of landscape and how national identity is perceived and experienced through ritual performance in the process of ancestral roots cultural heritage tourism, based on the study of the mausoleum.

## 3. Methods

### 3.1. Research Site: The Mausoleum of the Yellow Emperor

Fieldwork was conducted between 1 April 2015 and 15 December 2020 at the Mausoleum of the Yellow Emperor in Shaanxi Province, China. The Yellow Emperor is regarded as the ancestor of the Chinese nation. The mausoleum is the place where emperors and celebrities worship the Yellow Emperor (Figure 1). Meanwhile, the mausoleum has always been the place where successive dynasties held national sacrifices, preserving various cultural relics from the Han Dynasty to the present. As early as 1961, the mausoleum was announced by the State Council of China as the first batch of national key cultural relics protection units. In 2006, the Yellow Emperor Worshipping Ceremony (Yellow Emperor Memorial Ceremony) was included in the first batch of the national intangible cultural heritage list. In 2014, the mausoleum was included in the declaration of the world cultural heritage project.

On China's Qingming Festival (also known as Tomb Sweeping Day), visitors—including tourists from Hong Kong, Macau, and Taiwan—come to the mausoleum to worship the Yellow Emperor and attend the memorial ceremony. The layout and overall view of the mausoleum and memorial ceremony can be seen in Figure 2. Tomb Sweeping Day, also known as the Qingming Festival or the Ancestor Worshiping Festival, falls between mid-spring and late spring. It is the grandest ancestor worshiping festival for the Chinese nation. The ceremony has been upgraded from provincial level to national level and is among China's top five memorial ceremonies, as reported by the Xinhua News Agency in 2019; see Table 1. Moreover, the Yellow Emperor Memorial Ceremony ranks number one with regard to its long history and extensive influence, making it the most typical and representative intangible cultural heritage in China.

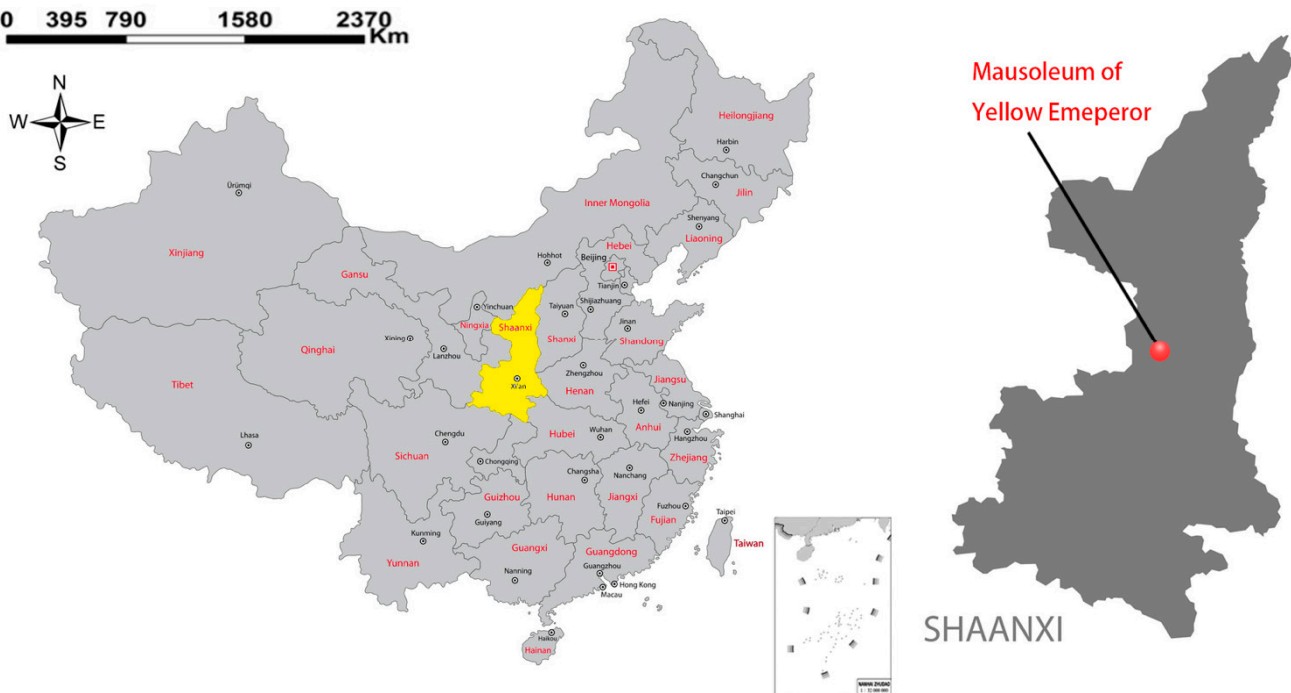

**Figure 1.** The Location of the Mausoleum of the Yellow Emperor (source: the authors).

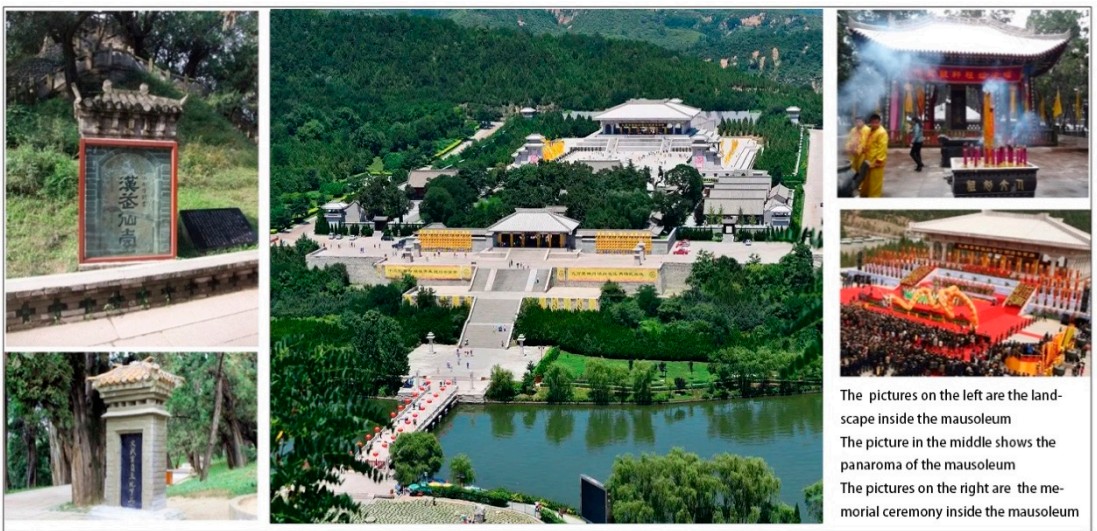

**Figure 2.** The Overall View of the Mausoleum of the Yellow Emperor. (source: the authors. Chinese characters on the top left picture refer to the stone tablet for worshiping; Chinese characters on bottom left picture refer to Dismounted Horse Stone. Chinese characters on the top right picture which are inscribed on the pavilion refer to the Yellow Emperor Memorial Ceremony. Chinese characters inscribed on the tripod refer to the ancestor of the Chinese nation).

**Table 1.** Top five memorial ceremonies in China (source: Xinhua News Agency).

| Ranking | Activity | Date of Ceremony | Historical Index | Impact Index | Participation Index |
|---|---|---|---|---|---|
| 1 | Public memorial ceremony in Shaanxi Province | Tomb Sweeping Day | 10 | 10 | 10 |
| 2 | Memorial ceremony to worship Yan Emperor in Hunan Province | Double Ninth Festival | 9 | 10 | 9.5 |
| 3 | Global ceremony to worship Confucius in Shandong Province | Mid-Autumn Festival | 10 | 9 | 9.5 |
| 4 | Yellow Emperor memorial ceremony in Henan Province | Lunar 3 March | 9 | 9 | 9 |
| 5 | Ancestor worship ceremony of Yandi in Hubei Province | Lunar 26 April | 8 | 9 | 9.5 |

*3.2. The Mausoleum of Yellow Emperor and Memorial Ceremony: Origin, History, and Changes*

3.2.1. Traditional Yellow Emperor Memorial Ceremony: Before 1949

From the Spring and Autumn period to the Warring States period, through the Qin and Han Dynasties and even the Qing Dynasty, the emperors of dynasties realized the legitimacy of political power and demonstrated their imperial power by worshipping heaven and the ancestors. The emperors of each dynasty conveyed the inherent relationship between their own imperial identity and the ancient emperor (the Yellow Emperor) through sacrificial words such as "obeying the order of heaven", "supporting the people by heaven", "building the pole after heaven" and so on, which helps to construct a hierarchical relationship between the monarch and the ordinary people.

As the ancestor of China, the Yellow Emperor was worshipped as the God of heaven. The emperors in ancient China established the authority of imperial power and expressed the wish to bless future generations through the worshipping ceremony of the Yellow Emperor, so as to consolidate their political power [96]. The transformation of the Yellow Emperor into political authority has become a symbolic resource attached and monopolized by the rulers of all dynasties. Worshipping the Yellow Emperor becomes a cultural symbol to highlight the legitimacy of the blood relationship and to consolidate imperial power. At present, the Yellow Emperor is no longer the exclusive ancestor of one family name but has become the common ancestor of the Chinese nation.

3.2.2. A local Ritual Ceremony: Between 1949 and 1978

During the War of Resistance against Japan, the Kuomintang and the Communist Party offered sacrifices to the Yellow Emperor and declared their strong determination to jointly resist foreign aggression. The Yellow Emperor was constructed as a "national common ancestor", and a unified national front line was established through this kind of cultural symbol. Since 1949, when the People's Republic of China was founded, the Yellow Emperor has become the "national ancestor" again. Over this period, the Shaanxi provincial government held a small-scale worshipping ceremony for the Yellow Emperor every year during the Tomb Sweeping Festival. The worshipping ceremony was mainly carried out at the provincial level. Representatives from the Shaanxi provincial government, the local county government, the local residents, and the media participated in the memorial ceremony.

3.2.3. The National-Level Ritual Ceremony: After 1978

The Reform and Opening-up policy in 1978 brought tremendous social, economic and cultural changes to the country. Large-scale renovations and expansions of the mausoleum were undertaken after 1978. The mausoleum was promoted as a national-level tourism

attraction, a sacred place, and a heritage tourism destination for visitors to worship Chinese ancestors and seek their ancestral roots. It is worth noting that Beijing adopted the peaceful approach in handling the cross-strait relationship and allowed for Taiwanese people to return to the mainland to find their "root" and worship their ancestor. The ancestor memorial ceremony at the Mausoleum of the Yellow Emperor began to attract large numbers of visitors from Hong Kong, Macau, and Taiwan, as well as overseas Chinese. They travel to the mainland in order to worship their Chinese ancestor—the Yellow Emperor. In so doing, the mausoleum worshipping ceremony has been a national-level event which exerts special influence on establishing connections between mainland China and Hong Kong, Macau, and Taiwan, as well as overseas Chinese societies.

### 3.3. Research Methods

In-depth interviews and archiving were employed as the main techniques of data collection. The interviews offer the inner thoughts and views of the respondents and were conducted between 2015 and 2020 during Tomb Sweeping Day and other public holidays (total fieldwork lasted 162 days). A researcher of this article applied for a permit to conduct fieldwork in the Mausoleum of the Yellow Emperor and received admission from the management center of the mausoleum. Interviews were conducted with the interviewees' consent in the public space of the mausoleum or in the places in which our interviewees felt comfortable. The interviewees were informed about the aim of this research and their right to leave the interview at any time before the interview. A total of 123 interviewees (54% female and 46% male) were recruited, including local residents, government officials, and tourists from different regions, not only from the mainland tourism market but also from Hong Kong, Macau, Taiwan, and other international markets (Table 2).

**Table 2.** Demographic characteristics of the interviewees (source: the authors).

| Demographic | Items | Frequency (N = 120) | % |
|---|---|---|---|
| Gender | Male | 65 | 54% |
| | Female | 55 | 46% |
| Age (years) | Under 18 | 19 | 16% |
| | 18–30 | 37 | 31% |
| | 31–40 | 22 | 18% |
| | 41–50 | 20 | 17% |
| | Above 50 | 22 | 18% |
| Education | Below high school | 23 | 27% |
| | Junior colleges | 40 | 28% |
| | Undergraduates | 48 | 38% |
| | Graduate or above | 9 | 8% |
| Occupation | Farmers | 20 | 22% |
| | Private business owner | 23 | 20% |
| | Government | 18 | 19% |
| | Education or research | 13 | 13% |
| | Companies | 17 | 14% |
| | Others | 19 | 12% |
| Degree of familiarity with the Yellow Emperor | Not at all | 5 | 4% |
| | A little | 39 | 33% |
| | Medium | 54 | 45% |
| | A lot | 21 | 18% |
| Regions | Tourists from Mainland China | 75 | 63% |
| | Tourists from Taiwan | 45 | 37% |

Snowball sampling was employed to recruit interviewees for the process when the research was conducted for the field work. In the snowball sampling, a group of respondents

was selected at the very beginning; they were usually randomly selected. After interviewing the first group of respondents, we asked them to provide other research subjects that met the research objectives. The next step was selecting the subsequent research objects in turn, according to the research objectives; the following research objects were selected in turn, thus forming a snowball effect. Each semi-structured interview lasted from 45 to 90 min and were all recorded. First, we contacted five local residents at the Mausoleum of the Yellow Emperor, and they introduced us to 35 other local residents in Huangling county for interviews. There were questions such as "how the memorial ceremony at the Mausoleum of Yellow Emperor would influence your daily life" and "How would you connect yourself with the Chinese nation". Because the residents' lives are repeatable and shared in the same community, new content failed to appear when we interviewed the 35th resident, which showed we had reached data saturation. The interviews of this part included 18 females and 17 males. We also tried to include groups of different ages. The oldest participant was 80 years old. The youngest participant was 19 years old. As the residents were quite familiar with the memorial ceremony, in this part of the interview we not only focused on the tracing back of the history of the mausoleum but also emphasized the emotions and attitudes of the residents towards the memorial ceremony. In addition, we also visited 48 government officials, including 30 males and 18 females. We first contacted three local government officials who worked in the Mausoleum of the Yellow Emperor, and they introduced us to 45 other government officials. Seventy-eight percent of the interviews with the government officials were taken at the Xuanyuan Temple as this was the specific site where the memorial ceremony was organized. The rest of the interviews with the government officials were taken at the Mausoleum of the Yellow Emperor as the government officials would worship the Yellow Emperor at the mausoleum after the memorial ceremony. The questions being asked in this part were those such as "what is the function of landscape and ceremony at the mausoleum" and "What are the considerations when naming and interpretating the meaning of landscape and ceremony". As the government officials are the major actors during the heritage tourism practice, in this part of the interview we therefore tried to explore the reasons behind the naming of the landscape and the meaning of the memorial ceremony. The last part of the interview was about the tourists, including 28 tourists from mainland China and 12 overseas Chinese. The interview questions included, but were not limited to, "how do you feel when you enter the Mausoleum?", "Would you rethink your connection with the Chinese state when you are here (the Mausoleum)", etc. The interviews with tourists were all taken at the mausoleum as it is a must site for the visitors, and we aimed to discover the tourists' perceptions and behavior towards the ancestral landscape and the memorial ceremony.

Archives contributed to the tracing of the history of the mausoleum and the commemorative ritual; the archives contain historical videos, photographs, books (i.e., local chronicles), local magazines, and newspapers. During our field work, we first categorized these texts, pictures, videos, stone tablets, and books, based on archival materials such as the Huangling County Chronicle, the Chronicle of the Mausoleum of the Yellow Emperor, the stone tablet collections of the mausoleum, the picture collections of the mausoleum, and other materials. Secondly, we classified these historical data in chronological order. Thirdly, we summarized the history of the Mausoleum of the Yellow Emperor as well the procedures of the memorial ceremony in different historical periods. Therefore, the archival materials are the sources that help us to sort out the origin, history, and changes of the Mausoleum of the Yellow Emperor and the memorial ceremony as well. Participant observation was employed to provide supplementary data. We observed the behaviors and emotions of the tour guides as well as the tourists. We paid attention to the tour guides' interpretations of different landscapes and how they guided the tourists to understand the different landscapes as well as the different procedures during the memorial ceremony. We observed how the tourists behaved when visiting the mausoleum as well as when attending the memorial ceremony. We observed what the differences were among the local

residents and the government officials, as well as the tourists in terms of the interpretation of the ancestral roots cultural landscape and the memorial ceremony experience.

Apart from three interviews which were regarded as invalid samples due to the low recording quality, a total of 120 interviews were found to be valid and were transcribed into text (in Chinese). All the interview transcripts were transcribed through an encoding process following the rule of "Ab1", in which A represents the initial letter of the place for the interview, B indicates the identity of the respondents (a represents government officials, b represents local villagers, c represents mainland tourists, d refers to overseas Chinese). For detailed analysis, themes were generated through a coding process based on the information gathered during the interviews. Relying on interviews, the researchers' aim was to understand the different groups' experiences and their perceptions of national identity at the mausoleum. Participatory techniques through active involvement with the guided tourists, friendly conversations, and evaluation of the written narratives were used to explore the relationship between the production of national identity and the heritage tourism practice [33]. Interrelated themes, such as the perceptions and emotions towards the Chinese nation were analyzed and explained to explore how national identity is produced and performed through landscape construction and memorial performance during heritage tourism.

Moreover, due to the influence of COVID-19, the Yellow Emperor Memorial Ceremony was carried out both online and offline. Travel restrictions after the pandemic made the field work impossible. Therefore, to determine the tourists' feelings and perceptions towards the Chinese nation during the Yellow Emperor Memorial Ceremony, we created a database from 16,006 visitor comments in 2020 from the online ancestor worship of the Yellow Emperor. The online worshipping ceremony for the Yellow Emperor attracted wide attention—not only people from the mainland but also Hong Kong, Macau, and Taiwan, as well as overseas Chinese. Messages about the "Online ancestor worship" were collected from the website of the Yellow Emperor Memorial Ceremony. Then, the messages were transcribed into R language software to extract keywords and calculate the frequency of keywords so as to visualize the public's opinion towards the commemorative ritual for the Yellow Emperor and their connection with the Chinese nation.

## 4. Landscape Naming and Practices of Nation-Building in the Mausoleum of the Yellow Emperor

### 4.1. Landscape Naming and Reinforcement of National Identity

Hall places the origin of the nation in a mythical narrative. Such myths not only contribute to the official narrations of the nation but also help to consolidate the political power [76]. The concept of Chinese culture is mainly related to traditional values and norms, especially the strong belief in ethnic and cultural homogeneity, which is the principle of the 'Descendants of the Yellow Emperor This principle is a mythical concept as old as the Chinese nation. Chineseness, in other words, means being Chinese. This concept perpetuates the Chinese beliefs in national and cultural homogeneity that stem from the mythical founder of the Chinese nation. The belief and pride in building a homogeneous nation plays a fundamental role in unifying Chinese culture and identity, especially in the face of national crises and foreign invasions. Through the ages the Chinese have claimed themselves to be the descendants of the Yellow Emperor, who was regarded as the 'Emperor of Heaven', especially in the ancient times. Therefore, worshipping the Yellow Emperor becomes an effective way to show the legitimacy and power of the nation.

When visitors come to see those landscapes, the representations of the landscapes play an important role in validating the visit [77]. The interpretation and symbolism of the landscapes helps in the understanding of their role in cultural practices. As the "capillary" of power, the landscapes can be carriers to show national identities [47]. The special relationship between national identity and cultural landscape can be addressed by observing the ways in which things and events are systematically brought together to represent the nation.

For instance, the tall and luxuriant cypress tree inside the mausoleum is said to have been planted by the Yellow Emperor himself. The cypress is constructed as the cultural symbol of the Chinese nation and national spirit through a semi-fictional means of establishing the connection between the physical elements and Chinese national history. The 95 steps leading up to the Xuanyuan Temple, called the Dragon Tail Road (*longwei dao*), symbolize imperial power. The naming of these landscapes helps to establish the links between the Yellow Emperor and Chinese history and culture [29]. Place names can be conceptualized as a form of heritage and the contemporary consumption of such names can be considered as a form of heritage tourism. A place name attached to a historic object is as meaningful to the visitors as the object itself [77]. The names of the landscape are represented as the connection between China, the Chinese, and the Chinese nation. They demonstrate that the nation constructs history in its own image to validate not only its existence but also its character and identity [97].

Dating back to the legendary period, the totem of the Yellow Emperor's tribe is the dragon and the Yellow Emperor himself was regarded as the incarnation of dragon. The symbol of the 'dragon' is the common element in the construction of memorial landscapes at the Mausoleum of the Yellow Emperor. Consequently, many sites are named after the symbol of the dragon. The scenic area's decorative sculpture, including the inscriptions and stone carvings, is mainly based on the dragon's image, such as the physical landscapes of the Dragon Tail Road, Dragon Soul Bell and Dragon Banner, as shown in Figures 3 and 4. According to a museum tour guide, the naming and interpretations of the landscapes are associated with the image of the dragon, which was based on the following.

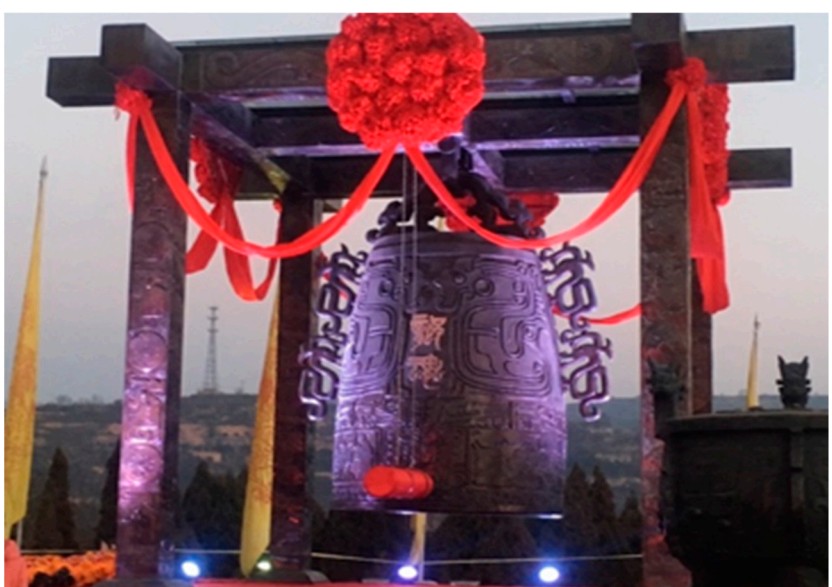

**Figure 3.** Dragon Soul Bell (source: the authors).

> *The temple courtyard has 95 steps, representing the Yellow Emperor's lofty status. Here we will see a variety of landscape images with the symbol of dragon. We are the descendants of the dragon. Like what we can see from this stone tablets sign by Deng Xiaoping, the founding father of the country. The four big characters 'descendants of the Yellow Emperor' are clearly displayed on the stone tablet in the pavilion of stone tablets.*

The symbol of the dragon in modern society is endowed with the connotation of the Chinese nation. It has become the cultural symbol of the Yellow Emperor's descendants. The names of the landscape embody the cultural connotation behind the physical existence itself. Moreover, the physical landscapes, such as the International Friendship Forest and the Chinese Century Cypress tree in the Mausoleum of the Yellow Emperor, have been endowed with new cultural connotations. The green and upright cypress trees become

the symbols of unity and harmony which represent the image of the Chinese nation [98]. The naming and interpretation of those landscapes is always interrelated with Chinese history and images. Those landscapes reflect the common history, collective memory, and common beliefs of the Chinese nation [99] constructing a unified and harmonious national image through the landscape naming [100]. This kind of historical and cultural symbol demonstrates the history and brilliance of the nation's past and forms the basis of the people's collective consciousness.

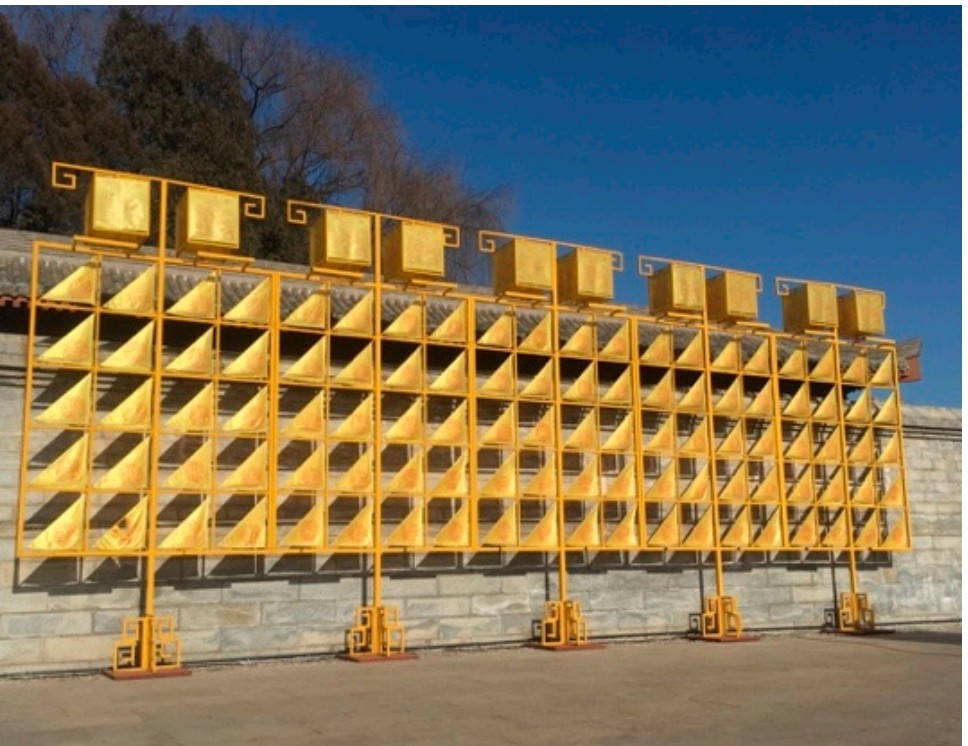

**Figure 4.** Dragon Banner (source: the authors).

*4.2. Ritual Practices and National Identity*

By emphasizing the common value of the society, ritual can enhance a sense of national belonging and national identity [101]. Ritual creates a sense of community through social and political life. The Yellow Emperor Memorial Ceremony was reinvented for heritage tourism, in which new symbols and traditions were endowed with new meanings and representations. For instance, the thirty-four drums in the ceremony are defined as the representation of the 34 provincial level political entities (including the provinces, autonomous regions, municipalities, and special administrative regions). The fifty-six dragon banners in the sacrifices were constructed as the symbols of the 56 nationalities of China, symbolizing the unity and harmony of the 56 ethnic groups in the nation. National image is defined as the representation that a person believes to be true about the nation and its people. According to Wang, national image is related to the opinions and perceptions of a country by its overseas populations [102]. An increasing number of countries attempt to improve their national image so as to expand their influence in the international arena. In the ritual ceremony, we not only mean the perceived image but also the projected image. The following interview with government officials in the local county confirms the meaning of the ritual practices.

> *Every year, when we offer sacrifices to our ancestor especially on the occasion of the memorial ceremony, we often prepare 34, because there are 34 provinces, autonomous regions, municipalities and special administrative regions under the central government. Fifty-six naturally means 56 ethnic groups, because the present memorial ceremony is*

*arranged by the province, and the political representatives of central government will also participate.*

The symbolic numbers, 34 and 56, are frequently used in the nation-building and identity-construction of China. Through grand ceremony and symbolic reconstruction, the strong, harmonious, unified, and prosperous image of the Chinese nation was "verified". The symbol of the dragon is also frequently used in the decoration of the ceremony, not only the sacrificial square but also the main avenue is decorated with dragon flags—including the image of a dragon printed on the yellow scarf worn during the ceremony. A flying dragon is also the key presence during the memorial ceremony, accompanied by singing and dancing, a 56-metre-long 'Chinese dragon' hovers, leaps, and rises into the sky. The local new report states that these ritual arrangements present the great rejuvenation of the Chinese nation:

> *There are nine dragon flags and nine phoenix flags respectively, implying the splendid nine dragon dynasty, nine phoenix flying, and the auspicious dragon and phoenix, symbolizing the solemn and powerful ceremony. 56 dragon flags mean that the Chinese people are the descendants of the dragon, and 56 nationalities jointly worship the Yellow Emperor as the ancestor of the Chinese nation.*

Both the local and national governments are instrumental in the organization and operation of the memorial ceremony. It has strict requirements, as mentioned in the interview with Mr. Wang, who is responsible for making offerings for the ceremony. Both the local and the national government try to embody the meaning of 'nation' and 'state' into the memorial ceremony so as to construct a unified and notable Chinese identity and a sense of belonging to the Chinese nation.

> *There is a strict limit on the number of offerings like 56 steamed buns for sacrifice on the ceremony, a symbol of the unity of 56 ethnic groups. This number is required by the county and the province. It can't be changed casually. One year, I changed the pattern and number, made 12 kinds of steamed buns with the appearance of Chinese zodiac, which were returned by the government and asked to be redone. Neither the appearance nor the quantity can be changed.*

The emotional entrainment is experienced as a feeling of belonging and togetherness, which is also instrumental in the development and maintenance of a collective consciousness. In this vein, political meanings, ideologies, and memories are inscribed on those symbolic entities, within which the state tries to establish its legitimacy and authority. The symbols achieve condensed representational efficacy by bringing together what scholars call an "ideological pole" and a "sensory pole" [103]. They can transform the intentions expressed in ritual practice into a transformative, determinable impact, thereby motivating individuals and groups to participate. Drums, bells, and flags can be tools to bring people together and strengthen their national consciousness and sense of belonging to China and the Chinese nation. The state impresses itself on the minds of its members through political doctrines, historical narratives, celebratory occasions, and memorial rituals [104]. Shared experiences and practices help to give new collectives an identity, while the figurative images convey messages associated with mythological figures [105].

As Hobsbawm has argued, symbols play a fundamental role in nation-building because they not only represent a sense of belonging but also embody the nation as an imaginary community [53]. Symbols visualize and emphasize national belonging by providing validity instead of the abstract concept of 'nation' [47]. National symbols are used to draw boundaries to assure a positive notion of national identity by emphasizing a nation's uniqueness and achievements. By carefully designing connotations and ideologies in the ritual process and practice, the Yellow Emperor Memorial Ceremony is embedded with specific political ambition.

## 5. Perception of the Nation and National Identity through Ritual Performance

*5.1. Ordinary Visitors' Experience and Perception*

Visits to place-name sights usually take place in heritage tourism. Place names are endowed with specific cultural associations, and the symbolic capital they represent becomes the focus of the tourist interest. Place names represent a distinctive lens through which to develop a deeper understanding of the production and consumption of heritage tourism [77]. The key to the Chinese nation is not its chronological history but its felt history [94]. The people's experience of heritage tourism can be considered as an important medium in maintaining the national imagination, especially through landscape naming and ritual performance. The union of landscape and ritual builds a material bridge for personal experience and national consciousness.

The Yellow Emperor Memorial Ceremony serves as the cultural practice to unify all the Chinese descendants. It also illustrates the identity of the Chinese and the Chinese nation. People claim to be 'the descendants of the Yellow Emperor' to show their strong emotions and attachment towards the Chinese nation and to express their national identity and pride. As for government officials, they are likely to present the great contributions of the Yellow Emperor through the development of a Chinese national culture as well as China as a political entity. This presentation by them is a unified and prosperous portrait of China, and the Chinese nation is constructed so as to consolidate national identity from the government's perspective. Mr. Tan, Vice Governor of Shaanxi Province, stated at the Ceremony of Public Sacrifice in 1980:

> *Qingming is a national festival to memorize the ancestors. The Yellow Emperor is the common ancestor for all Chinese, as descendants of China and the Yellow Emperor, no matter where we are, all workers, peasants, intellectuals, people in Taiwan, overseas Chinese and all people should contribute to the great rejuvenation of the Chinese nation and the reunification of the motherland.*

There have been official attempts to construct, represent, and promote national identity through sanctifying national figures and ideologies from the legitimate position—we can find similar discourses in the interviews with local government officials. Mr. Su, a local government official, still remembers the worshipping ceremony towards the Yellow Emperor when he was a child:

> *When I was a child, my parents would bring me to worship our ancestor on Tomb Sweeping Day. At that time, the happiest thing was to get some sacrificial offerings after the worshipping ceremony, which was regarded as the luckiest thing. As the native resident, growing up here in Huangling, this is the place like our home where all our emotion and affection belongs to. When we attend the ceremony, we can see people from all walks of life, and we all share the same past. We are all parts of the descendants of [the] Yellow Emperor because we share one common ancestor.*

In recent years, on-line mourning and memorialization have become more and more popular in China, especially in the post-pandemic era. The management center of the heritage set up a website for netizens to memorize during Tomb Sweeping Day. As presented in Figure 4, the main attribute of national identity is the belief that the Chinese are the descendants of the Yellow Emperor. As Park has argued, this belief is understood as the core value of the nation in the consciousness of the Chinese people [13]. Accordingly, heritage is not only a kind of physical entity, it is also a symbolic embodiment of the past, constantly reconstructed and reconstituted in the collective memory [106]. The common emotional attachment to the Yellow Emperor indicates the personal importance of Chinese cultural homogeneity and its coherent symbolic manifestations as key elements of national identification. The emotional attachment of tourists and local residents to their ancestor was expressed in spontaneous and unconscious ways. Tourists and local residents enhance their sense of national belonging by sharing and exchanging memories of the nation. It has been emphasized that national identity is closely associated with emotional attachments to the nation rather than its outward manifestations.

As for the non-local tourists, they place their focus more on the culture of the Yellow Emperor, which is also viewed as the root of national culture and Chinese civilization. This kind of culture also exerts great influence on tourists nowadays. As Figure 5 shows, the Yellow Emperor is highly recognized as the ancestor of the Chinese nation. The Yellow Emperor is always associated with the symbol of 'China', 'Motherland', and the 'Chinese Nation', just as Mr. Tang mentioned in the following interview.

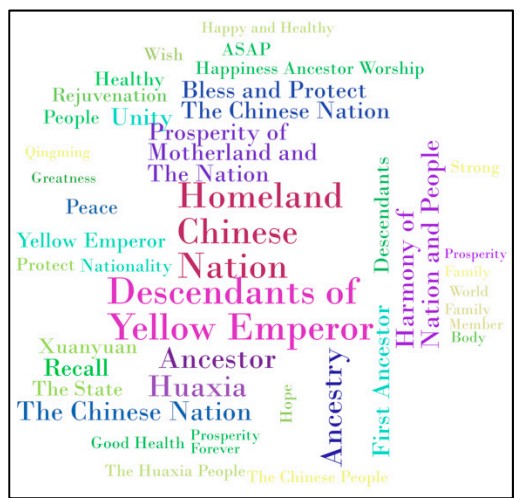 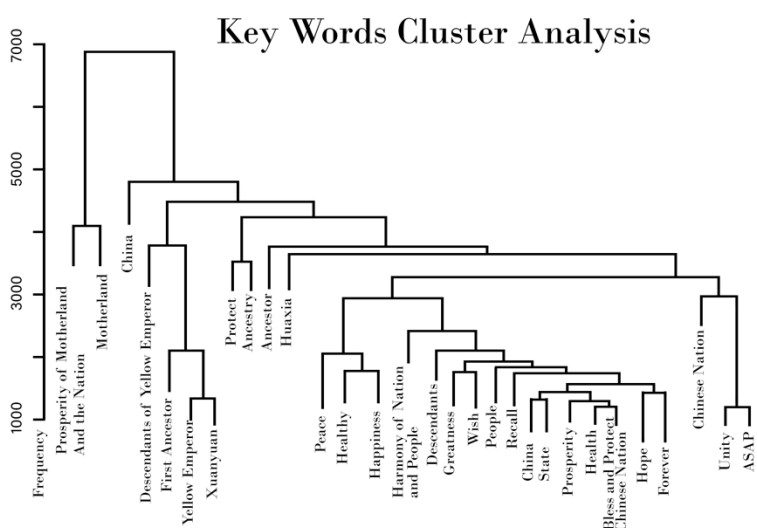

**Figure 5.** Keyword visualization map of the Yellow Emperor Memorial Ceremony (source: the authors).

> *I have lived in the UK for 17 years and my hometown is in Shaanxi Province. This time I bring my families to attend the ceremony. The ceremony was magnificent and grand. When we are at the site of the mausoleum, when we hear the ancient music, listen to the past history and suffering of the Chinese nation with other compatriots, the memorial ceremony reminds us our past life in China. We feel that we are not alone, wherever we go, we always belong to the Chinese nation. We have the same ancestor and we have our roots here. We are connected with our motherland by blood tie, no one can cut it off (Interview with overseas Chinese tourists).*

Visits from political leaders also make the heritage tourism much more popular. Tourists believe that worshiping the Yellow Emperor and attending the memorial ceremony are not only the processes of praying for blessings and wishes for family and friends, but also a way to follow the path of famous political figures. Political leaders promote political meaning, ideologies, and national identity through their discourse and narrative during heritage tourism. 'Blessing China' and 'May the country be prosperous and the people at peace!' are two phrases used to show people's gratitude towards the ancestor and their best wishes for the Chinese nation. Here, they can recall the memory of the past and obtain spiritual comfort—especially the elderly tourists. Heritage tourism not only represents history but also the discourse practice of memory and imagination in the past [106]. People express their wishes and expectations for the motherland through the Yellow Emperor Memorial Ceremony.

> *The memorial ceremony was quiet spectacular. Many people came including political representatives from the national government. I heard that China's top leaders had also come before. As senior citizens, our lives are quite good now compared with the past. I remember in the past, there were not so many magnificent pavilions and grand buildings here. Thanks to our old ancestor, thanks to the communist party of China and our country, praying for our ancestor to protect our happy life. We are all descendants of the Yellow Emperor; I am proud to be part of the Chinese nation. I hope the ancestor will bless and protect us. We wish our country becomes stronger and more prosperous.*

Toponyms represented a particular form of symbolic capital with a powerful appeal to a specific group of tourists [77]. The Yellow Emperor and the ceremony still impose an influence on tourists. The memorial ceremony serves as the catalyst that ignites the emotions and affections of tourists towards the Chinese nation. The ritual practice and performances of tourists play an important role in reproducing the meaning of tourist spaces and places [107]; the activities and 'practices of place-named tourists' are instrumental in reconstituting the symbolic meaning and the significance of particular toponyms and re-sacralizing them as 'attractions'. Personalities are widely constructed through practices of consumption [77]. For those tourists, visiting and worshipping at the Mausoleum of the Yellow Emperor is not only a way to show their recognition for Chinese culture but also their emotional attachment to China and the Chinese nation. Therefore, cultural, national, and place identities are entangled through visiting and worshipping. For local residents, the Yellow Emperor and the mausoleum play an important role in their daily lives. The local county is constructed as the hometown of the dragon, which is considered a holy land for the Chinese nation. This embedding of the meaning of a place helps to attract more visitors to the mausoleum to worship the Yellow Emperor. Because of the development of tourism and the promotion of the memorial ceremony, the local county has been transformed from a remote village into a famous heritage tourism destination and the holy land for the Chinese nation. Therefore, the local residents are very proud of their hometown and strongly recognize the culture of the Yellow Emperor.

*5.2. Taiwanese Visitors' Experience and Perception*

Since the 1980s, the Yellow Emperor Memorial Ceremony has been the important connection between Chinese people, including those living outside of the mainland. In 1987, the Chinese State Council issued *Suggestion on Taiwanese visiting family members in mainland China* to facilitate the Taiwanese visiting their family members in mainland China. Later, an increasing number of Taiwanese family groups returned home and visited the mainland. Instead of directly returning home and visiting family members, some of them first went and worshiped at the Mausoleum of the Yellow Emperor. For the Chinese state, the Yellow Emperor and its related Chinese nationality legitimizes the connection between the mainland and Taiwan. The Chinese state promotes the Mausoleum of the Yellow Emperor as the root for all Chinese descendants (*Yan Huang Zisun*), which represents a common cultural basis between Chinese people and their ethno-cultural identity. For the Chinese state, promoting the ancestral roots culture could contribute to promoting the national identity of China among the Taiwanese. This is exemplified in the following interview with students from Tsinghua University (Taiwan):

> *Our first history class is about the story of the Yellow Emperor. We know that we are the descendants of the Yellow Emperor, but we only hear stories from books. The memorial ceremony left us with a deep impression. We were quite excited to attend the ceremony. When wearing the Yellow sacrificial kerchief, which symbolizes the descendants of the Chinese people, there is a sense of coming back to the hometown. When I came here I knew where my nation came from and how the blood was spread. We have the common origin and ancestor. There are only some differences in our accents; as long as we communicate, there are no cultural barriers and difficulties. After going back, I will share my experience with my friends. I hope they will have the chance to come here to find their roots and worship their ancestors.*

For Taiwanese tourists attending the worshiping ceremony, phrases such as 'motherland', 'coming home', and 'homesickness' make a great impression on them. The devout ancestor worshiping performance helps to sacralize the attractions during heritage tourism, which plays a crucial role in reproducing the meaning of the attractions as well as the tourist space. Tourist activities and performances are instrumental in reconstructing the symbolic meaning and significance of the place and space [107]. Here, mainland China is likely to be the motherland and home where Taiwan represents the son who left the motherland. This kinship metaphor suggests the Taiwanese tourists' emotional attachment

to mainland China through the Yellow Emperor Memorial Ceremony. These emotional attachments to mainland China are common among the Taiwanese tourists who participate in this ceremony, as the Taiwan tourist states in an interview which can be seen in the following part.

*I was told in 1984, when the first Taiwanese came back to participate in the worshiping activity, their clothes displayed phrases such as 'xiang nian zuguo' ('missing motherland'), 'zuzong de erzi' ('son of motherland'), and 'zhongyuhuijiale' ('finally coming home'). They were very devoted and excited. I was impressed. When I attended the memorial ceremony myself, I strongly felt we are part of the national culture and we are the descendants of the Yellow Emperor.*

The above informant frequently travels between Taiwan and the mainland, which enhances his recognition of the ancestor, the Yellow Emperor. Here, his personal identity is woven with national identity. However, these types of intertwined factors are mostly revealed from the interviewees who migrate from mainland China to Taiwan. Another Taiwan tourist articulates in an interview that,

*My father left the mainland for Taiwan on 15 March 1949. This is my first time to participate in this worshiping ceremony. I think I should come because it is our Chinese's ancestor. I have read about the culture of the Yellow Emperor in the history textbooks before. This time, while watching the performance of drum and bell ringing, I was moved by the respectful reading of sacrificial articles, music and dancing performance in the ceremony. I think I will come again.*

Moreover, the heritage of the mausoleum is a geopolitical approach used to unify all the Chinese people—including the visitors from Taiwan—through the naming of landscapes and ritual practices. Notably, in 2015, the representatives from Taiwan who attended the ceremony also participated in the tree-planting activities. The government highlights its ideological and political goals of unifying the Taiwanese through the naming of the landscapes. The cypress forests had been named the Rising Dragon Friendship Forest and the Forest for the Descendants of the Yellow Emperor, respectively, as shown in Figure 6. When place names are endowed with specific cultural associations, the symbolic capital they represent becomes the focus of tourist interest. This kind of place-named tourism usually takes place within the broader practices of heritage tourism and cultural tourism. Place names can be visited and consumed for their extraordinary associations or meanings. Place names as attractions are also distinctive in semiotic terms. The name (marker) is of more significance than the sight to which it refers [77]. They all signify that Taiwan and mainland China share the same roots and ancestry. The feelings of respect and gratitude for the Yellow Emperor were expressed through tree-planting activities.

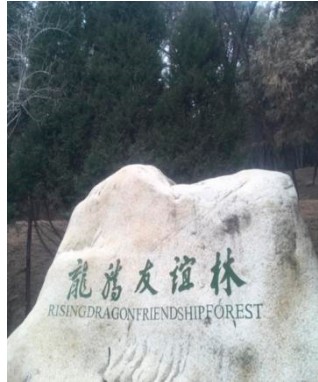 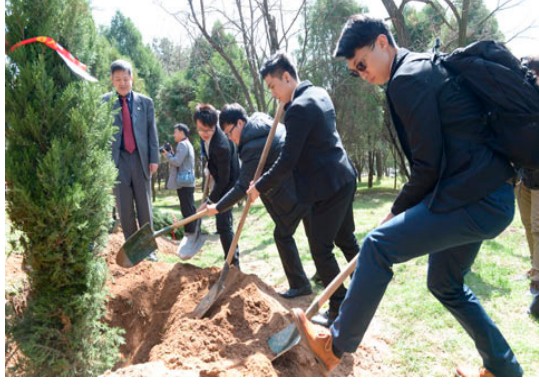

**Figure 6.** Symbolic landscape: cypress forests at the Mausoleum of the Yellow Emperor (source: the authors).

Numerous cypresses were planted in the mausoleum by the political representatives in memory of the ancestor. In order to express respect and gratitude for the ancestor, a cypress forest named Siyuanlin was planted in 2011 by the political leaders to signify the same cultural roots between mainland China and Taiwan. Literally, it implies that Taiwan and mainland China share the same ancestor, culture, and identity. Place names may generate particular expectations among visitors. A place can be transformed and commodified to meet the expectations of visitors. It is clear that for many visitors the encounter with such a place is often a significant and meaningful experience, as the names of places and landscapes are often associated with iconic figures or events from a kind of typical culture. The visit can be an occasion to be engaged in acts and remembrance. Thus, what the tourists 'do' at such places is far from trivial; instead, they are engaged in purposeful acts of meaning-making [77]. Similar sentiment is stated by a representative of the Taiwanese visitor who participated in the tree-planting activity.

> *During Tomb Sweeping Day, the descendants of Yellow Emperor gathered at Qiaoshan at home and abroad, worshiped the Yellow Emperor, and planted the cypress in Qiaoshan, which will help to enhance the unity of compatriots across the Straits, and further deepening friendship and cohesion. The cypress planted today will surely thrive and become a towering tree. Instead of worshiping our ancestor, it will silently express our sincere feelings and deep homesickness. Tree-planting activities for ancestors have enhanced the cohesion of the Chinese nation. We cherish the friendship between our compatriots from all over the world and their blood ties. We should continue to publicize, inherit and carry forward the Chinese civilization created by the Chinese people's ancestor.*

The descendant of the Yellow Emperor has perpetuated the Chinese belief in ethnic and cultural homogeneity, and thus, the mythological founder of the Chinese nation. As the common ancestor for the Chinese, the Yellow Emperor is recognized by both sides of the Taiwan Straits. People show strong recognition for the culture of the Yellow Emperor along with the national culture. In short, the Mausoleum of the Yellow Emperor and its memorial ceremony represent a kind of national symbol for nation-building, and by constructing the close association between the Yellow Emperor and Chineseness, the Chinese government promotes political dialogue and harmony between the two sides of the Taiwan strait.

## 6. Conclusions and Discussions

Many battle-related sites and memorial sites focus on discussing the relationship between identity and heritage motivations. They find that a positive narrative of history helps to connect the individuals' emotions and feelings with the heritage site. The heritage site acted as a tangible reminder to enhance the collective self-esteem [108,109]. Later, more theories (like Urry's tourist gaze and MacCannell's 'modern tourist') were used to explain the battle-related tourism experience. Not only war sites but also amusement sites, heritage sites, and cultural sites arouse the researcher's attention [110]. In the examination of the role of heritage tourism in producing national identity, this paper argues that the Mausoleum of the Yellow Emperor stands as a visual and symbolic testimony for sustaining national solidarity and legitimacy. We try to figure out how national identity is produced through heritage tourism practices. From the perspective of the invention of tradition, we find that the government plays a very significant role in the production of the ancestral roots cultural heritage site. The naming and interpretation of landscapes and the ritual performance in heritage tourism are closely associated with national history and mythology. Specifically, the mausoleum and its ceremony are mainly embodied as symbols and icons of China and the Chinese nation. Heritage tourism serves as the tool to draw closer ties among the Chinese, especially between mainland China and Taiwan. For the Taiwanese visitors, worshiping the Yellow Emperor demonstrates their associations with the Chinese culture and the mainland. Visiting the Mausoleum of the Yellow Emperor helps to enhance their understanding of Chinese culture. Behind the scenes, we find the political mechanism enacted by the Chinese government highlighting the notion of the Chinese and China as a

powerful and unified nation through the naming and interpretation of the ancestral roots cultural landscape, as well as through the ritual practices.

This paper tries to address the relationship between heritage tourism and national identity, focusing on discussing the intermediating roles of heritage tourism in establishing and promoting identity construction. We emphasized the sociopsychological dimension and the emotional significance of heritage as shared memories and values. It also reveals that the cultural politics of landscape are inseparable from the economic relationship. Our research not only focuses on discussing the meanings and representations of physical landscapes but also emphasizes the exploration of the social construction process and the power relations behind landscape naming and ritual performance. What the landscape is does not matter, what really matters should be how it works as an approach to expressing national and cultural identity. This paper points out that landscape and ritual act as a carrier for proclaiming the cultural authority and as a producer of interests for different stakeholders [111,112]. We find that the practice of the landscape naming and ritual performance in heritage tourism is closely related to the development of the Chinese nation.

We also find out that the heritage site and its symbolic significance serve as a tangible reminder to reaffirm the national meanings and the attached values. The process of heritage tourism not only promotes national economic development but also develops the cultural symbol for China and the Chinese nation. The appeal to tourists of particular toponyms can be used by tourism planning and marketing agencies as well as by souvenir providers in the process of place branding and marketing projects. This paper encourages local communities to utilize the symbolic capital of their names and make contributions to local economic development through place-named tourism [62,77]. This research not only helps the individuals to realize and reappreciate the value of traditional culture and heritage but also motivates the individuals to rethink their responsibilities in cultural inheritance and the innovative development of culture. We also demonstrate that positive historical narratives and engaged heritage tourism experience helps to enhance national cohesion. This research is also conducive to the psychological and emotional connection of the Chinese descendants of immigrants from all over the world so that people can share the same destiny and help to enhance the national identity as well as the cultural identity.

This research has at least two major limitations. First, as mentioned in the method section, the fieldwork was conducted from 2015 to 2020, which was before the outbreak of the COVID-19 pandemic. Due to travel restrictions and city lockdowns during the pandemic, further fieldwork and information are currently unavailable. Second, as tourism has largely been at a standstill during the pandemic, most local strategies and practices of landscape naming and ritual performance in the mausoleum are paused. The nuanced politics among the different stakeholders deserves further exploration in future studies. The negotiation and power relations among different stakeholders needs to be explored to show the different motivations of different stakeholders. Place names may generate particular expectations among different visitors that do not match the local community itself, thereby disappointing the visitors in their experiences. These kinds of particular issues need further research in the future.

**Author Contributions:** Conceptualization, H.W. and Z.Y.; Data Curation, H.W.; Methodology, Y.Y.; writing—original draft preparation, H.W.; writing—review and editing, Z.Y. and H.W.; funding acquisition, Y.Y. All authors have read and agreed to the published version of the manuscript.

**Funding:** This paper was supported by the funding from Major Projects of the National Social Science Foundation entitled "Research on the Integrated Development" of "Red Culture + Tourism" in Old Revolutionary Areas Grant number (21&ZD179).

**Institutional Review Board Statement:** Not applicable.

**Informed Consent Statement:** Informed consent was obtained from all subjects involved in the study.

**Data Availability Statement:** Not Applicable.

**Acknowledgments:** The authors wish to acknowledge their sincere thanks to the editors and reviewers whose insightful guidance and valuable suggestions helped a lot in improving the quality of the paper. Special thanks to Bo Zhang and Su Feng for their kind advice and constructive suggestions during writing.

**Conflicts of Interest:** The authors declare no conflict of interest.

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
