# Peer review of "Heritage Tourism and Nation-Building: Politics of the Production of Chinese National Identity at the Mausoleum of Yellow Emperor"

_sustainability, doi:10.3390/su14148798_

Round 1
Reviewer 1 Report
This paper discusses heritage tourism as a specific symbolic tool through which the national identity is constructed which is, indeed, a fascinating topic. The study utilizes various approaches, including in-depth interviews and secondary sources analysis which is a laudable combination. I like the paper; it is interesting, well-structured, and original.
However, there are some points I want authors to reconsider, modify, or add. I believe it will improve the whole quality of this academic paper.
The main concern for me is a discussion regarding the naming, both theoretical and in the study results. It's the weakest segment, frankly.
Firstly, from what I read in the manuscript, I cannot get real connections in the analyzed literature between the 'cultural politics of landscape naming" and "ritual practices of heritage tourism." The authors try to connect these concepts using providing statements without the real support from the previous studies. For example, the authors claim (line 222): "It is expected that tangible and physical elements of heritage such as the establishment and naming of statues, squares, and monuments will contribute to the construction of national identity." Great, but how do you know that such a connection exists? What about the supporting points/pieces of evidence regarding the place names and monuments as the sources for national identity construction? It may be true, but I cannot find any examples in your theorizations, and it sounds just like some speculative theorizations. At the same time, there are excellent works that might support your points, the works that, in fact, are difficult to avoid discussing the questions of political toponymic, commemoration, and heritage. They must be analyzed in the theorization of the problem and added to the list:
Palonen, E. The city-text in post-communist Budapest: street names, memorials, and the politics of commemoration. GeoJournal 73, 219–230 (2008). https://doi.org/10.1007/s10708-008-9204-2
Light, D. (2014) Tourism and toponymy: Commodifying and Consuming Place Names, Tourism Geographies, 16(1), 141-156. DOI: 10.1080/14616688.2013.868031
Fuchs, S. (2015). History and heritage of two Midwestern towns: a toponymic-material approach, Journal of Historical Geography, Volume 48, 11-25, https://doi.org/10.1016/j.jhg.2015.01.003.
Moreover, there are no real connections between the rituals and place names in the toponymic literature the authors provided (e.g., line 245 with Brenda Yeoh 1996 reference – it is about toponymy and nation-building, right, but not, actually, about rituals). The name of the segment 2.3 Cultural politics of landscape naming and ritual practices in heritage tourism implies close relationships, but I cannot see them from the text.
Furthermore, what kind of naming do you have in your work? The naming of what kind of landscape? Cultural? Symbolic? Urban? I cannot get it from the discussion. Basically, the theorizations we have in place naming are primarily based on urban streetscapes (with only some exceptions). What names (the names of what) do precisely the authors want to discuss? It should be explained in detail.
Next, what kind of place names are you going to discuss? Naming "squares" – this is about toponyms/place names, but naming "rituals" – it is not about toponymy/place naming; it is about the process – different categories in onomastics (the study of names). The naming of places is not the same as the naming of rituals. For example, see the author's notion on lines 243 -244: "Indeed, names of landscapes and rituals are not only geographical markers but also symbols of national memories and ideologies." This is a purely speculative statement because I cannot see the theorization about the rituals as "geographical markers" (it may be true, but I cannot see this from the theoretical discussion; instead, I see some unsupported by academic works statements).
Additionally, a set of data (a list of names though you have just a couple of them – which is not a problem as there are many excellent works in political place naming based on even one place-name) must be provided (as a table, for instance).
Importantly, I fail to get connections to the theoretical segment 2.3. in part 5. "Landscape naming and practices of nation-building in the Mausoleum of the Yellow Emperor." In this discussion, the authors follow some other approaches far from those stated in their theorizations. There are no relations to the critical place naming scholarship the authors discussed. What is interesting there the author decided to separate "naming" and "rituals" (subsections 5.1 and 5.2). And naming is now discussed in connection to nation-building. What was the reason for discussing them together in theory? Again, the authors are talking about "landscape naming" there – what kind of landscape?
To be more to the point, I suggest rewriting these segments (2.3. and part 5). Also, the abstract and the conclusions should be modified based on these changes. It affects all the paper, its structure, and its logic.
Some other (technical) points:
The language of the manuscript should be polished.
Please double-check references – there are some mistakes (e. g. 888 – must be "toponymies," not "toponymics")
All text must be double-checked thoroughly (e.g., Line 48 – references family name to fix; Line 388 – "a tall," etc.)
I hope my suggestion will be helpful.
Author Response
Dear editor and reviewers,
We appreciate the opportunity to revise our manuscript entitled “Heritage tourism and nation building: politics of the production of Chinese national identity at the Mausoleum of the Yellow Emperor,” and we thank you for taking the time to provide such insightful guidance.
We carefully considered your comments as well as those offered by the three qualified reviewers. Below, you will find revisions and replies to the comments made in response to the manuscript. We assigned numbers to the comments (in italics) so that we could appropriately address and cross-reference each specific issue/comment/question. Responses are below each comment in regular (non-bold) font. Within the responses, italicized sections denote changes we made to the manuscript itself. Within the manuscript itself, changes are highlighted in red. Where we made pervasive or exceptionally lengthy changes, we ask that you please refer to the manuscript itself. Please see the attachment. We thank you for your efforts in enhancing the quality of the manuscript and hope that the below revisions/responses satisfactorily address your concerns.
Thank you again for your effort and time. We look forward to hearing from you soon regarding this revised paper.
Sincerely,
The Authors

Reviewer 2 Report
I appreciate the opportunity to review the intriguing manuscript. The relationships and connections between landscape, rituals, and national identity are investigated in this study. Based on a case study conducted at the Mausoleum of the 'Yellow Emperor' in Shaanxi, China. It investigates how China (as a nation) and Chinese (as a national identity) is represented, performed, and constructed through landscape interpretation and visitors' ritual experience in heritage tourism, which is an innovative topic. However, some concerns must be addressed during the subsequent revision.
The interview questions should be included in the "Methods" section so that readers can better comprehend the research process and logic.
In addition, there is no description of the procedure for dealing with the interview content, as well as the interview's findings and discussion.
Part four devotes a considerable amount of space to the description of The Yellow Emperor and its worshiping ceremony: origin, history, and changes, as well as Landscape naming and nation-building practices in the Mausoleum of the Yellow Emperor. In addition, what is the significance of this section, and how does it contribute to the study's conclusion?
The theoretical contribution requires additional discussion in the discussion section. What theoretical contributions have been made to heritage tourism and national identity?
The practical implications have not been elaborated upon in detail. Does this study have any implications for regional government or heritage tourism professionals? What are the ramifications for mainland tourists and Chinese living abroad?
There is no description of the future directions of research. The author should also discuss how to address the limitations of the research, as well as potential future topics that could be addressed.
Author Response

(The authors gave the same response as above.)

Reviewer 3 Report
This manuscript offers an interesting look at the relationship between heritage tourism and national identity from a cultural geography perspective. With some revisions, I think it could make a valuable contribution to the literature.
Most importantly, the purpose of the research is unclear. In the abstract, the authors state that the “paper examines how the Chinese national identity is produced, performed, and established in both physical and symbolic approaches.” The authors further note that the “paper discusses emotional bond for casting the consciousness of the Chinese nation’s community as well as the theoretical contribution for the sustainable development of local community in the post-pandemic era.” While the latter part of this statement pertaining to sustainable development in the post-pandemic era seems designed to appeal to the theme of the journal and the special issue, discussion of this topic is almost nonexistent in the manuscript. Further, the authors state in the introduction, “This paper explores the relationships and connections between landscape and rituals and national identity”. Clarifying the purpose of the research, and organizing the paper around that stated purpose, will strengthen the manuscript.
The literature review is extensive, with a combination of original theoretical contributions and recent perspectives; however, I would like to see the authors add some discussion regarding the application of the concepts from the Western-oriented literature to the Chinese context.
The methods section could benefit from additional information to allow the reader to understand and possibly replicate the procedures undertaken. It would be helpful to know more about the recruitment process for interviewees, how long the interviews lasted, when the interviews took place, if there were any significant differences in interviews over the time period (especially during COVID), etc. Significantly more information is needed about the use of participant observation, archival materials, and online comments, including what, exactly was used (e.g., what was observed on tours – guides, tour content, tourist behaviors, what online forums were used, etc.) how the use of these materials contributed to the purpose of the research, and how they were analyzed. It is unclear how these different sources are used in the findings section as well. Most of the comments seem to be from interviews. One comment is attributed to a tour guide; was this in an interview or observed on a tour? The comments on page 14 are not attributed to a particular source. I would also suggest the authors provide more context on “tomb sweeping day” for international readers.
Without a clear purpose, it is hard to know what the authors want the reader to take from the conclusion. What are the theoretical and/or practical implications of this research?
Overall, the manuscript is clearly written and organized, with only minor grammatical and typographical errors that can be addressed with a thorough editing. The figures are generally appropriate. The insert photo of the Mausoleum of the Yellow Emperor on the map is small. The reader might benefit from a larger image of the site. The significance of the first two images in Figure 5 are not readily apparent to international readers, nor does the caption adequately explain it.
Author Response

(The authors gave the same response as above.)

Round 2
Reviewer 1 Report
I would like to thank the authors for their hard work with the manuscript. Indeed, many elements were modified, adjusted, and the paper looks much more reliable for a reputable international journal.
I would also like to point out that there are some notable issues with the modified segments which diminish the quality of the paper. Therefore, I would recommend the authors to follow the comments and fix these elements.
1. In the abstract there is no mention of place naming. There is a substantial theoretical discussion about toponyms, and the authors have discussed place names in their results of the study. What is the reason for this if there is not even one point about the place names in the abstract?
2. I have some questions regarding the modified section 2.3.
2.1. Lines 195 – 196: “Alderman, Azaryahu and Rose-redwood as the pioneers, they began to focus on the political, cultural and symbolic meaning of place names (Ji 2016).” Actually, all the pioneers can be found in Berg and Vuolteenaho’s (eds.) book (2009), including Vuolteenaho and Berg who coined the term “critical toponymy” (they must be cited immediately when the authors used this term for the first time - it seems line 195).
3. Line 230 – Should it be critical toponymy? What does it mean by “critical toponym’? How does it differ from “uncritical”? This is an odd term. I also fail to get the logic of the sentence.
4. The term “abundant” (line 253). If you have examples of such “abundant” scholarship in rituals and naming of landscapes, please provide the list of sources after this sentence.
5. Line 907 - place-named tourism – Light (2014) must be cited.
6. The sentences in the modified segment should be polished for English and grammar (e.g. Rose-Redwood). Sometimes it is really difficult to follow (see, e.g., line 197: “How the political awareness and social values influence the naming of places also arouse the attention from the researchers.” Or some even unfinished sentences – line 212-213; also there are many mistakes and typos (e.g., line 251). I believe this problem can be easily fixed (though the first that catches the readers’ attention, and is not necessarily a native one).
7. There is a serious issue with the reference list. Some authors did not mention (e.g., Light 2014, the key work based on your modified manuscript!) The structure also looks unsystematic (see, e.g. -ref. 78 – 80, lines 1069 - 1073). Some information is often omitted in the source (see, e.g. missed year of publication in ref. 96., line 1100). I fail to get the reference style as well - 2008,9(4): 431-452 for ref. 95 vs. 34,453-470 for ref. 96. Some references have the wrong publishers (e.g. ref. 109 must have Chicago, Aldine Publishing Company, not Cornell University Press; ref. 88 – Oxford does not relate to Routledge; ref. 80 – Oxford does not relate to Ashgate, etc.) This is an academic paper in the international journal, and it should be referenced accordingly.
I hope these recommendations can be useful.
Author Response
Dear editor and reviewers,
We appreciate the opportunity to revise our manuscript entitled “Heritage tourism and nation building: politics of the production of Chinese national identity at the Mausoleum of the Yellow Emperor,” and we thank you for taking the time to provide such insightful guidance.
We carefully studied reviewer’s comments point by point and have made revisions which marked with different colours in revised the manuscript accordingly. Below, you will find revisions and replies to the comments made in response to the manuscript. Please see the attachment. We assigned numbers to the comments (in italics) so that we could appropriately address and cross-reference each specific issue/comment/question. Responses are below each comment in regular (non-bold) font. Within the responses, italicized sections denote changes we made to the manuscript itself. Within the manuscript itself, the amendments for the first round are highlighted in red, changes for the second round are highlighted in blue. Where we made pervasive or exceptionally lengthy changes, we ask that you please refer to the manuscript itself. All authors have approved the response letter and the revised version of the manuscript. We thank you for your efforts in enhancing the quality of the manuscript and hope that the below revisions/responses satisfactorily address your concerns.
We would like to express our sincere thanks to the reviewer for the constructive and positive comments. So much appreciated for your effort and time. We look forward to hearing from you soon in due course.
Sincerely,
The Authors

Reviewer 2 Report
Thank you for your excellent revision work. Overall, I am content with the revision and author's responses. Regarding the random sampling procedure, one minor point must be clarified: how is random sampling carried out?
Except for the above minor revision suggestion, I believe the revised manuscript is ready for publication in its current form.
Author Response
Dear editor and reviewers,
We appreciate the opportunity to revise our manuscript entitled “Heritage tourism and nation-building: politics of the production of Chinese national identity at the Mausoleum of the Yellow Emperor,” and we thank you for taking the time to provide such insightful guidance.
We carefully studied reviewer’s comments point by point and have made revision which marked with different colours in revised the manuscript accordingly. Below, you will find revisions and replies to the comments made in response to the manuscript. Please see the attachment.We assigned numbers to the comments (in italics) so that we could appropriately address and cross-reference each specific issue/comment/question. Responses are below each comment in regular (non-bold) font. Within the responses, italicized sections denote changes we made to the manuscript itself. Within the manuscript itself, the amendments for the first round are highlighted in red. changes for the second round are highlighted in blue. Where we made pervasive or exceptionally lengthy changes, we ask that you please refer to the manuscript itself. We thank you for your efforts in enhancing the quality of the manuscript and hope that the below revisions/responses satisfactorily address your concerns.
We would like to express our sincere thanks to the reviewers for the constructive and positive comments. We look forward to hearing from you soon in due course regarding the revised manuscript.
Sincerely,
The Authors

Reviewer 3 Report
The authors have strengthened the manuscript with their revisions. The beginning of the abstract in particular offers a clearer direction for the manuscript. However, the statement that the “paper tries to encourage younger and older generations to realize and reappreciate their intimate and inseparable connections to the Chinese nation and its people, with particular reference to emotional and sociopsychological elements of heritage” requires further clarification. How, exactly, will the paper accomplish this?
In addition, the statement about sustainable development of local communities still seems out of place and unsupported in the manuscript (primarily one sentence in the conclusion). This should be further developed or removed from the abstract.
Figure 1 is clear and easy to read without the small inset photo of the mausoleum. Can the authors include a higher resolution photo of the mausoleum as a separate figure to give the reader an image of the site?
Section 3.3 Research Methods is improved with the additional information regarding interview procedures. It is still unclear how the archival materials were used in this study.
Author Response
Dear editor and reviewers,
We appreciate the opportunity to revise our manuscript entitled “Heritage tourism and nation-building: politics of the production of Chinese national identity at the Mausoleum of the Yellow Emperor,” and we thank you for taking the time to provide such insightful guidance.
We carefully studied reviewer’s comments point by point and have made revision which marked with different colours in revised the manuscript accordingly. Below, you will find revisions and replies to the comments made in response to the manuscript.Please see the attachment. We assigned numbers to the comments (in italics) so that we could appropriately address and cross-reference each specific issue/comment/question. Responses are below each comment in regular (non-bold) font. Within the responses, italicized sections denote changes we made to the manuscript itself. Within the manuscript itself, changes for the first round are highlighted in red. Changes for the second round are highlighted in blue. Where we made pervasive or exceptionally lengthy changes, we ask that you please refer to the manuscript itself. We thank you for your efforts in enhancing the quality of the manuscript and hope that the below revisions/responses satisfactorily address your concerns.
We would like to express our sincere thanks to the reviewers for the constructive and positive comments. We look forward to hearing from you soon in due course regarding this revised paper.
Sincerely,
The Authors
